# T-REX: The tile-based representation of lateral exchange processes in ICON-Land

Philipp de Vrese<sup>1</sup>, Tobias Stacke<sup>1</sup>, Veronika Gayler<sup>1</sup>, Helena Bergstedt<sup>2</sup>, Clemens von Baeckmann<sup>2</sup>, Melanie Thurner<sup>3</sup>, Christian Beer<sup>4,5</sup>, and Victor Brovkin<sup>1,5</sup>

**Correspondence:** Philipp de Vrese (philipp.de-vrese@mpimet.mpg.de)

Abstract. The vast majority of land-surface models uses a tiling-approach to capture the effects of subgrid-scale spatial heterogeneity in the land-surface properties. In most cases, however, the tiles, which represent patches with homogeneous characteristics at and below the surface, are treated independently of each other and the lateral exchange between them is not being taken into consideration. The present manuscript describes an approach for a tile-based representation of lateral exchange processes in heterogeneous landscapes that was recently implemented into ICON-Land, the land surface component of the ICON framework. The scheme captures the horizontal fluxes on a broad range of spatial scales and represents 5 lateral exchange processes, namely gravity-driven moisture fluxes and the corresponding convective heat transport, diffusive- and conductive fluxes of water and heat as well as the wind-driven redistribution of snow. In the approach, the relationships between any two tiles are determined by a set of characteristic connectivities which are treated as inherent properties of the pair of tiles – invariant in time and independent of the location – and derive from the internal logic underlying the definition of the tiles. The characteristic connectivities are used to calculate the spatio-geometric relationships between two tiles, such as the (geometrical) contact length or the characteristic center-to-center distance between two adjacent surface clusters. These, in turn, define gradients as well as the time-lag factors that govern the lateral transport in the model. In addition to a description of the model development, we present two example applications of the new scheme, which address the effect of sub-grid scale fluxes on the model's ability to capture the spatial variability in the state of the surface and sub-surface and the overall terrestrial water storage. Here, our results suggest that lateral exchange processes, especially on small horizontal scales, are highly relevant for the spatial variability in the soil temperatures and for the simulated extent of surface water bodies, while the effects on the grid-cell mean state and the turbulent exchange with the atmosphere appear to be largely negligible.

<sup>&</sup>lt;sup>1</sup>Max Planck Institute for Meteorology, Climate Dynamics, Hamburg, 20146, Germany

<sup>&</sup>lt;sup>2</sup>b.geos, Korneuburg, 2100, Austria

<sup>&</sup>lt;sup>3</sup>Karlsruhe Institute of Technology, Institute of Meteorology and Climate Research - Atmospheric Environmental Research, Garmisch-Partenkirchen, 82467, Germany

<sup>&</sup>lt;sup>4</sup>Universität Hamburg, Department of Earth System Sciences, Hamburg, 20146, Germany

<sup>&</sup>lt;sup>5</sup>Universität Hamburg, Center for Earth System Research and Sustainability (CEN), Hamburg, 20146, Germany

#### 1 Introduction

20

Most land surface models (LSM), especially those included in Earth system models, have been designed to run with horizontal grid spacings of tens to hundreds of kilometers. While such resolutions might cover an important part of the spatial variability in the atmospheric state variables, this is not necessarily the case for soil- and land-surface properties which can vary at the (sub-) meter scale. Here, the coarse resolution may entail aggregation errors and hinders the representation of distinct ecosystem states which shape the physical and biochemical processes at and below the surface (Rastetter et al., 1992; Beer, 2016). Even with the grid spacing of some applications approaching the kilometer-scale (Stevens et al., 2019; Bogenschutz et al., 2023; Lee and Hohenegger, 2024), many of the relevant processes can still not be resolved. To deal with the issue, most models employ a tiling-approach (Avissar and Pielke, 1989; Koster and Suarez, 1992). This approach is based on the assumption that the land surface within one grid cell can be split into sub-divisions, which each comprise patches with homogeneous characteristics, and that model processes can be determined for each of these sub-units, the tiles, separately. Most often, the individual tiles do not interact directly, but only through the exchange with the overlying atmosphere (Molod et al., 2003; Best et al., 2004; de Vrese et al., 2016b; Huang et al., 2022), allowing for a computationally inexpensive treatment of the subgrid-scale variability since the landscape heterogeneity does not need to be be resolved in a spatially explicit way.

Land cover constitutes a key determinant of the local energy and water cycles, with vegetation affecting virtually all terms of the surface energy balance and the moisture exchange with the atmosphere (Seneviratne et al., 2010; Duveiller et al., 2018). Topography is another important factor of heterogeneity and induces variability on all scales. The microtopography often determines the position of the local water table and soil temperatures, which, in turn, control soil respiration rates and their sensitive to fluctuations in the controlling variables (Sommerkorn, 2008). At the other end of the scale-range, topographic ridge-to-valley gradients are the main cause for a moisture convergence at the hillslope- and catchment-scale (Fan et al., 2019). And while topography and land cover arguably constitute the more prominent factors, soil properties may also have a strong impact on the local state of the surface and subsurface. Soil structural properties are closely linked to infiltration rates, the water holding capacity (Bordoloi et al., 2018; Basset et al., 2023) and the soil thermophysical properties (Zhang and Wang, 2017). Here, horizontal gradients in the soil organic matter content, in particular, have a large potential for inducing sub-grid-scale variability in soil moisture as well as in surface and below-ground temperatures (Rawls et al., 2004; Koven et al., 2009; Fekete et al., 2012; Chadburn et al., 2015; Langer et al., 2016; Zhu et al., 2019; Siewert et al., 2021; Zhang et al., 2021).

With land cover being one of the most important sources of heterogeneity, the tiles have traditionally been used to represent different vegetation classes or plant functional types. These are key determinants of the properties shaping the physical land-atmosphere interactions – albedo, roughness and surface resistance to evapotranspiration – as well as for the processes governing the terrestrial carbon cycle. However, studies have shown that, depending on the region and processes considered, other factors may be as important as the vegetation cover, with one of the most prominent examples being agricultural irrigation (de Rosnay et al., 2003; Boucher et al., 2004; Sacks et al., 2009; Guimberteau et al., 2011; Puma and Cook, 2010; de Vrese

et al., 2016a; de Vrese and Hagemann, 2017; Chou et al., 2018; Singh et al., 2018; Hauser et al., 2019; Cook et al., 2020; Al-Yaari et al., 2022; McDermid et al., 2023). Here, the soil moisture in well-confined areas is maintained at high levels often in (semi-) arid regions and several models employ tiles to distinguish between managed and unmanaged agricultural areas (Yao et al., 2025).

Since the tiles possess distinct properties, they also develop distinct hydrological and thermophysical states, e.g. with average surface temperature differences between irrigated and non-irrigated grid-cell fractions and between grass- and tree tiles being in the order of degrees (Schultz et al., 2016; Thiery et al., 2017). In reality, numerous processes act to both reduce and increase horizontal gradients in the state of the surface and the subsurface, with most models (implicitly) making extreme assumptions on the effectiveness of these processes. As stated above, the most prominent assumption is that there is no direct interaction between the tiles, other than through the exchange with the atmosphere. The other extreme assumption, which is used e.g. in JSBACH3 the land surface component of the Max Planck Institute for Meteorology's- and the AWI Earth System Model, is that the lateral exchange is highly effective so that the physical state variables, most importantly soil moisture as well as soil and surface temperatures, are continuously averaged between tiles (Reick et al., 2021). It is obvious that these assumptions are only valid for either extremely large or extremely small horizontal scales but misrepresent a large fraction of the real-world landscape heterogeneity. A broadly applicable representation of subgrid-scale heterogeneity requires an explicit representation of the lateral exchange between tiles and many advances have been made to incorporate such interactions, especially with respect to the hydrological fluxes (Aas et al., 2019; Fan et al., 2019; Swenson et al., 2019; Blyth et al., 2021; Chaney et al., 2021; Smith et al., 2022; de Vrese et al., 2024; Li et al., 2024).

In the following, we describe a recently developed scheme for including horizontal subgrid-scale fluxes of water and heat in ICON-Land, the land surface component used within the ICON modelling framework (Jungclaus et al., 2022); namely a tile-based representation of lateral exchange processes in heterogeneous landscapes (T-REX). First, we give a concise overview of the assumptions and the governing equations of the scheme (sec. 2), before investigating the effects that the lateral exchange processes have on the simulated state of the surface and subsurface, using two example applications (sec. 3). Here, the first application targets the effect of lateral subgrid-scale fluxes on the terrestrial water storage as well as the resulting impact on the turbulent land-atmosphere exchange and the state of the surface (sec. 3.1). With the second example, we investigate the effect of the lateral heat transport on the spatial subgrid-scale variability in the soil temperatures, focusing on typical patterned-ground structures commonly found in periglacial regions (sec. 3.2).

#### 2 T-REX

T-REX is designed to enable simulations in which the prevalent spatial variability in the state variables and the exchange fluxes with the atmosphere can be captured by the tiling scheme. Thus, the model needs to represent the interactions between all those tiles that are required to resolve the subgrid-scale heterogeneity in the determining factors of the local hydrological


conditions and the thermophysical state at and below the land surface. In reality, the spatial variability comprises a wide range of length-scales (Blöschl and Sivapalan, 1995; Seyfried and Wilcox, 1995), requiring the model to determine the lateral transport processes for a similarly broad range of scales (Fig. 1): On the micro scale, moisture conditions can vary dramatically across distances of a few meters, which can also entail pronounced temperature gradients, especially if there is a strong effect on the soil thermophysical properties – that is heat conductivity and capacity – and on the energy released by or required for phase changes in the soil (Langer et al., 2011). Macro-scale moisture convergence – that is on length scales of hundreds of meters to a few kilometers – may result in pronounced vegetation gradients and landscape types such as gallery forests and raised bogs (Li et al., 2024), with a large potential of macro-scale soil moisture differences to notably affect temperatures via their effect on the evaporative cooling (Thiery et al., 2017).

Figure 1. Spatial subgrid-scale heterogeneity

Shown is an idealized representation of the spatial heterogeneity that is not resolved by the horizontal grid and how this is treated by the tiling scheme of the model. Here, the land surface may exhibit heterogeneity with respect to a number of characteristics, most importantly differences in topography, land cover or soil properties. Small scale features (here, represented by the model by the tiles 2 and 4), that is patches or clusters with horizontal length scale of a few meters to tenths of meters, interact with the larger features they are encompassed in (tiles 1 and 3). Depending on the kind of heterogeneity, non-negligible interactions can also persist across distances of hundreds of meters to a few kilometers, that is between the surface clusters represented by tile 1 and tile 3. The model does not treat the clusters or patches individually but aggregates them into tiles, that are represented by a cover fraction and by a set of characteristic properties. The state of the clusters and the lateral exchange between them is then determined based on the characteristic properties that describe the tiles themselves and also the connections between each pair of tiles.

T-REX includes 5 lateral exchange processes that can be categorized with respect to the transported quantity (water or heat), the scales the processes are relevant for – that is the micro-scale (horizontal distances of roughly 1 m to 100 m) and the macro-scale (roughly 100 m to 10000 m) – and the mechanism driving the transport (gravity-driven-, wind-driven- and con-



ductive/diffusive fluxes) (Fig. 2). On the micro scale the scheme represents the diffusive- and conductive fluxes of water and heat (secs. 2.3.1 and 2.4.1) as well as a wind-driven redistribution of snow (sec. 2.3.3). Furthermore, the model accounts for gravity-driven fluxes of water, which we simply refer to as surface- and subsurface runoff (sec. 2.3.2), and the corresponding convective heat transport (sec. 2.4.2). A diffusive and conductive macro-scale exchange is not included in the model, with gradients being negligible due to the large distances considered. On this scale, the lateral transport is limited to the gravity-driven fluxes of water and the resulting convective heat fluxes. In general, the representation of the transport processes is based on a set of factors that describe the degree to which two tiles are connected (sec. 2.1) in combination with a number of explicit or implicit assumptions on the spatial relationship between the tiles (sec. 2.2). In our technical implementation, the interaction between two tiles is performed sequentially. This involves computing fluxes and gradients between an individual tile, t, and an equivalent connected tile, which we refer to as a sibling tile, s.

Figure 2. Lateral transport processes

Overview of the five lateral transport processes that are included in T-REX, grouped by the transported quantity, the underlying mechanisms and the scales at which the exchange is assumed to have a notable effect on the physical state of the land surface.

Finally, we would like to highlight that numerous approaches exist to describe any of the processes that the model represents. Some are more simplistic than others and they may also vary substantially with respect to their computational costs and data requirements. For the present implementation, one of our main goals was to design the parametrizations as consistent as possible with the assumptions that ICON-Land's soil-hydrology- and thermophysics routines are based on, since the lateral exchange processes are tightly coupled to the vertical movement of water and energy. Here, however, T-REX should be thought of less as a final suit of parametrizations of lateral exchange processes and more of a framework that allows for a consistent treatment of the inter-tile exchange, with enough flexibility to facilitate the implementation of improved parametrizations and new processes.





#### 2.1 Connectivities

The main assumption of T-REX is that the relationships between any two tiles can be described by a set of characteristic connectivities. These are used to calculate the spatial relationships between two tiles (see sec. 2.2) as well as the time-lag factors (see sec. 2.3.2) that govern the lateral transport. In the present implementation, they are treated as inherent properties of the pair of tiles, invariant in time and independent of the location and the composition of a grid cell. Here, the characteristic connectivities merely describe the general connection between two types of surface clusters, based on the internal logic underlying the tile-definition of a given setup. However, they do not necessarily constitute the degree to which two tiles are connected within a specific grid cell. Since in most cases, the intertile-exchange does ultimately depend on the grid-cell composition, the actual connectivities also account for the fractional cover of the tiles within a given grid cell (see below). The characteristic connectivities  $c_p^{t,s}$  between a tile t and its siblings s need to be provided to the model in the form of a  $n \times n$  connection matrix, where n is the number of tiles considered in the setup. T-REX uses several matrices to describe the lateral interactions between tiles, since the connectives depend on the process p that is being considered.

The most basic of these matrices provides the characteristic connectivities due to direct contact between two tiles. Here, the matrix elements  $(c_{ctc}^{t,s})$  constitute the relative (geometrical) contact lengths, corresponding to the fraction of the boundary of a surface patch of tile t that interfaces with a patch of given sibling tile s. These connectivities are used to determine the spatial relationships between a tile and its siblings and all those lateral fluxes that are directly proportional to the inter-tile gradient in the corresponding state variables – that is the conductive and diffusive fluxes of heat and water – as well as the redistribution of snow. The matrix elements need to be provided in such a way that i) if a connectivity  $c_{ctc}^{t,s} > 0.0$  is assumed between t and t, a connectivity t and t and

As stated above, the standard setup does not use the characteristic connectivities  $c_{ctc}^{t,s}$  directly, but, in order to account for the composition of a grid cell (making the assumption of a well-mixed distribution of tiles) they are weighted by the cover fraction of a given sibling tile ( $fc^s$ ) relative to the cover fractions of all connected siblings:

$$c_{ctc*}^{t,s} = \frac{c_{ctc}^{t,s} \cdot fc^s}{\sum_{k=1}^{nct} c_{ctc}^{t,k} \cdot fc^i} \cdot \sum_{k=1}^{nct} c_{ctc}^{t,k}.$$
(1)




The above is, however, an optional formulation and in cases that the definition of the tiles dictates that the connectivity is, in fact, independent of the grid cell composition,  $c_{ctc*}^{t,s}$  can also be set to  $c_{ctc}^{t,s}$ .

With respect to surface- and subsurface runoff and the corresponding convective heat fluxes, it is assumed, that the movement of water does not happen uniformly across the surface or through the soil, but through dominant hydrological flow paths. Consequently, the runoff fluxes from a tile to its connected siblings may not be directly proportional to the contact lengths. A good example for this are tiles that are connected to two siblings with different surface elevations. Here, the relative contact length may be the same with respect to the two siblings but water at the surface would only run off to the sibling that has a lower surface elevation than the tile. The connectivities are distinct for the micro- and macro-scale fluxes as well as for the surface and subsurface flow. Consequently, the present T-REX setup considers four additional matrices, representing micro-scale connectivities at the surface  $(c_{hfp,micro,srf}^{t,s})$  and below ground  $(c_{hfp,micro,blg}^{t,s})$  as well as macro-scale connectivities at the surface  $(c_{hfp,macro,srf}^{t,s})$  and below ground  $(c_{hfp,macro,blg}^{t,s})$ . On the macro-scale, connections may even exist for two tiles that do not interface directly, but where the hydrological flow paths are assumed to pass the area of another tile without interacting with the latter. This simplification was required to be able to represent runoff across hillslopes - where water may initially pool in micro-scale depressions – with only a small number of tiles. Hillslope runoff may best be described by a depressional fill-spill cascade, in which a fraction of the runoff initially pools in small scale depressions until it overflows and moves down-slope across the even surface of the slope to again pool in a lower lying depression (McDonnell et al., 2021). For many applications, it may, however, not be feasible to resolve the lateral fluxes representing individual height bands, as proposed by e.g. Chaney et al. (2021), and the entire hillslope is subdivided into two tiles, representing the relatively even slope surface - that is the areas where water may not pool – and small scale depressions. In such a setup, the model can not represent hill-slope flow cascades and the inclusion of a flux from the depression tile back to the tile representing the even surface fraction results in a circular connection that has the potential to retain water on the slope indefinitely. To avoid such closed loops, the runoff from the depression tile can be routed to a connected lowland tile directly, bypassing the tile representing the relatively even surface areas of the slope.

#### 2.2 Spatial relationships

On the macro scale, all spatial relationships between connected tiles are merely implicitly included in the parametrizations of the model, via the slopes and the time-lag factors used to determine the macro-scale runoff (see sec. 2.3.2). On the micro scale, however, the lateral exchange between a tile and its connected sibling tiles is in large parts determined by spatial relationships that the model determines explicitly, more specifically by differences in surface elevation, the distance between the centers of the patches that the tiles represent and the characteristic interface length of the connection. These are determined based on the characteristic surface elevation, the characteristic area covered by a patch of a given tile, the fraction of the area of the patch that is typically connected to a patch of a specific sibling-tile and by the cover fractions of the tiles, all of which need to be provided as input parameters.

# 2.2.1 Horizontal relationships

Figure 3. Assumed horizontal relationships between connected tiles

The top row shows the most basic case of land-surface heterogeneity, in which the patches of an inner tile are randomly distributed and fully contained within the patches of an outer tile. The second row shows a configuration in which the inner tile is connected to two outer tiles, with different cover fractions but approximately the same connectivity to the inner tile (resulting in the same interface length but different distances between the centers of the tiles). The third row shows a 2-level nesting, where a middle tile contains an inner tile and is connected to two outer tiles with different cover fractions as well as a different connectivity with the middle tile (resulting in roughly the same center-to-center distance with the middle tile). With the patches of the inner tile fully contained by the patches of the middle tile, the assumed relation between these two tiles is the same as between those shown in the top row. The bottom row shows a n-level nesting (n > 2), in which the horizontal relationships for each pair are determined based exclusively on the characteristics of the two adjacent tiles.

The scheme assumes a constellation in which one of two (micro-scale-) connected tiles, the outer tile  $(t^o)$ , encompasses the second, the inner tile  $(t^i)$ , at least partially (Fig. 2). Furthermore, it is assumed that  $t^i$  represents well distributed subgrid patches, the shape of which can be approximated by a circle  $(C^i)$  with the characteristic area  $A^i$ . This allows the calculation of the characteristic interface length between tiles  $t^i$  and  $t^o$ ,  $t^{i,o}$ , as the circular arc whose fraction of  $t^i$ 's circumference is equal to the relative contact length  $t^i$ .

$$l^{i,o} = (A^i \cdot \pi \cdot 4)^{0.5} \cdot c_{ctc*}^{i,o}. \tag{2}$$

 $C^i$  forms the inner region of a second circle  $(C^a)$  which contains the characteristic areas of  $t^i$  and of  $t^o$ , and whose radius  $(r^a)$  is used to approximate the center-to-center distance between the two tiles  $(d^{i,o})$ . The latter is assumed to be equal to half the distance between the circumferences of  $C^i$  and  $C^a$  added to  $r^i$ , the radius of  $C^i$ . The area of  $C^a$  is determined so that a vector with a central angle  $\Theta$  (in radians) equal to the relative contact length contains an area equal to  $A^o$ , the characteristic area of the encompassing tile  $t^o$ , and the area of a vector of  $C^i$  with the central angle  $\Theta = c^{i,o}_{ctc*}$ . To ensure the consistency between the assumed circular shape of the patches that compose  $t^i$  and the cover fractions of the two tiles, the scheme does not use the characteristic area of the outer tile directly, but an approximation  $(A^o)$  that is based on the characteristic area of the inner tile  $(A^i)$  and on the cover fractions of the outer  $(fc^o)$  and the inner tile  $(fc^i)$ .

$$\begin{split} d^{i,o} &= r^i + \frac{r^a}{2} & \text{with} \\ r^i &= \left(\frac{A^i}{\pi}\right)^{0.5} & \text{and} \\ r^a &= \left(\frac{\frac{A^o_i}{c^i \cdot c^s} + A^i}{\pi}\right)^{0.5} & \text{and} \\ A^o_* &= A^i \cdot \left(\frac{fc^i + fc^o}{fc^i} - 1\right). \end{split}$$




In case of a multilevel, nested connectivity – that is a constellation with the encompassed tile surrounding a third tile – the calculation of  $l^{m,o}$  and  $d^{m,o}$  between the middle  $(t^m)$  and the outer tile  $(t^o)$ , are based on the characteristic area  $(A^i)$  of the innermost tile  $(t^i)$ . Here, the current implementation of the scheme only allows for a two-level nesting – i.e.  $t^m$  may only contain one tile  $t^i$ , which itself does not encompass another tile – in which  $t^i$  is fully surrounded by  $t^m$   $(c^{i,m}_{ctc*}=1)$ . In such cases,  $l^{m,o}$  is calculated by:

$$l^{m,o} = (A_*^c \cdot \pi \cdot 4)^{0.5} \cdot c_{ctc*}^{m,o}, \tag{4}$$

where,  $A_*^c$  constitutes the combined areas of the inner and of the middle tile and is calculated based on the same assumptions used to determine  $A_*^o$  in eq. 3.

$$A_*^c = A^i \cdot \left(\frac{fc^i + fc^m}{fc^i}\right). \tag{5}$$

 $d^{m,o}$  is determined analogously to eq. 3, with the difference being that  $A^c_*$  is used instead of  $A^i$  and that  $A^o_*$  now accounts for the cover fractions of all three tiles:

$$d^{m,o} = \frac{r^i + r^a}{2}$$
 with 
$$r^i = \left(\frac{A^i}{\pi}\right)^{0.5}$$
 and 
$$r^a = \left(\frac{\frac{A^o_*}{c^{m,o}_{cte*}} + A^c_*}{\pi}\right)^{0.5}$$
 and 
$$A^o_* = A^i \cdot \left(\frac{fc^i + fc^m + fc^o}{fc^i} - 1\right).$$

It is possible to use the model for higher levels of nesting, however, in these cases the horizontal relation between two tiles is determined for each pair of tiles individually using equations 2 and 3, which means that a different shape is assumed for each tile depending on whether it constitutes the inner tile or the outer tile in the connection to a given sibling-tile. With the exception of the innermost tile in the hierarchy, this, however results in an underestimation of the interface-length and an overestimation of the center-to-center distance since it assumes the inner tile to have the shape of a circle rather than that of a ring.

## 2.2.2 Vertical relationship

The horizontal moisture fluxes and the associated heat transport between two adjacent tiles also depend on the assumed vertical relation between the two. Here, the most important parameter is the difference in surface elevation ( $\Delta h^{t,s}$ ) between a tile (t) and its connected sibling-tile (s). There are several possibilities to account for difference in surface elevation in the parametrization of the lateral movement of water, with the most straight-forward being the consideration of the geodetic head gradient in the Richards equation. However, since gravity-driven and suction-driven fluxes are being treated separately, T-REX follows a different approach.

Concerning the gravity-driven fluxes, the difference in surface elevation are used to determine the micro-scale slope  $(m_{mic}^{t,s})$  between two tiles:

$$m_{mic}^{t,s} = \frac{\Delta h^{t,s}}{d^{t,s}}$$
 with 
$$\Delta h^{t,s} = h^t - h^s.$$
 (7)

The micro-scale slopes, in turn, are used to derive the effective slope of a given tile  $(m^t)$  which determines the amount of runoff generated under a given level of saturation of the soil (see sec. 2.3.2). The effective slope is approximated by the sum of the macro-scale slope  $(m^t_{mac})$ , which is an input parameter for the model, and the weighted sum of all sibling-specific micro-scale slopes:




$$m^{t} = m_{mac}^{t} + \sum_{s=1}^{nct} m_{mic}^{t,s} \cdot c_{ctc+}^{t,s} + m_{mic}^{s,t} \cdot c_{ctc+}^{s,t}.$$
 (8)

Here, the sibling-specific slopes are weighted using a modified connectivity  $c_{ctc+}^{t,s}$  ( $c_{ctc+}^{s,t}$ ) which is based on the interface lengths ( $l^{t,s}$ ) and only accounts for those connections with a positive (negative) offset in surface elevation:

$$c_{ctc+}^{t,s} = \begin{cases} \frac{l^{t,s}}{\sum_{k=1}^{nct} l^{t,k} + l^{k,t}} & \text{if } \Delta h^{t,s} > 0.\\ 0 & \text{otherwise} \end{cases}$$
 and 
$$c_{ctc+}^{s,t} = \begin{cases} \frac{l^{s,t}}{\sum_{k=1}^{nct} l^{t,k} + l^{k,t}} & \text{if } \Delta h^{t,s} 

Figure 4. Refined vertical grid

Shown is the vertical grid used by the model to determine the diffusive and convective fluxes between two (micro-) connected tiles. This refined grid  $z^*$  encompasses all the layers of the vertical grid on the tile  $z^t$  and the layers fn the vertical grid on the sibling tile  $z^s$  shifted by the height offset between the two.

The ingress of water  $(D_{i*}^{d,e})$  into a section of the soil of a relative elevation (e) that is located within the depth of the surface water body of a relative depression (d) is calculated in the same way, using a matric potential of zero for standing water at the surface. Since the model does not contain any information on the relative location of surface water bodies, it is assumed that these are distributed evenly within the surface area of a given tile. Consequently, the relative contact length  $l^{d,e}$  is reduced according to the maximum inundated fraction of the depression  $f_{p,max}^d$ .

$$D_{i*}^{d,e} = f_{p,max}^d \cdot l^{d,e} \cdot \Delta z_{i*} \cdot k_{i*}^e \frac{\psi_{i*}^e}{d^{d,e}}.$$
 (11)

It should be noted that the model does not simulate the lateral egress of water from layers that are located above the surface of the water body as these fluxes are accounted for in the formulation of suburface runoff (see sec. 2.3.2).

# 2.3.2 Surface and subsurface runoff

# **Runoff generation**



In the standard configuration, ICON-Land uses the ARNO-rainfall—runoff model to determine the partitioning of precipitation (P), into infiltration (I) and runoff (R) (Todini, 1996; Dümenil and Todini, 1992; Reick et al., 2021). The scheme has been designed for applications at the watershed-scale – that is horizontal grid spacings >  $10 \,\mathrm{km}$  (Blöschl and Sivapalan, 1995) – and assumes variations in topography to induce a sub-grid-scale soil moisture distribution which, leads to spatially non-uniform infiltration and runoff rates. The use of the ARNO-model becomes problematic if the resolution of the model increases to a

point when grid cells no longer represent entire catchments or drainage basins, since the assumptions on the sub-grid-scale soil moisture distribution may no longer be valid. Here, an increase in resolution may not only refer to the actual grid-spacing of the model but also to setups in which the land surface model is applied to represent specific sites, implying a plot-scale resolution of the model. In case that T-REX is applied to tiles representing certain subsystems of a catchment, the ARNO-scheme can also not be used to determine the tile-specific generation of surface and subsurface runoff. For such cases a set of alternative formulations has recently been implemented in the model, which are based on a point-scale approach and make no assumptions about any sub-grid scale variability or spatial extent of the area represented by the model (for more details see appendix A).




In the point scale approach, surface runoff  $R^t_{srf}$  from a given tile (t) occurs if the infiltratable water – that is the sum of rain (or throughfall in case of a vegetation cover), snowmelt, water stored in the surface reservoir and runon from connected tiles – exceeds the soil's infiltration capacity  $(I^t_{cap})$  and the depression storage capacity  $(I^t_{dpr})$ . Here, the model distinguishes between the water that may infiltrate across the entirety of the surface area  $(I^t_{pot,sl})$  and the water that may additionally infiltrate below inundated areas  $(I^t_{pot,pnd})$ , with  $fc^t_{pnd}$  being the inundated fraction). With respect to the infiltration capacity, the model assumes the latter to be equal to the saturated hydraulic conductivity of the uppermost soil layer, reduced according to the layer's ice content. Runoff also occurs if infiltration results in super saturated soil layers, i.e. the infiltrated water can not percolate downwards fast enough or drain from the soil either at the soil bedrock interface or laterally from the soil layers above the bedrock boundary:


$$\begin{split} R^t_{srf} &= R^t_{srf,h} + R^t_{srf,d} & \text{with} & (12) \\ R^t_{srf,h} &= R^t_{srf,sl} + R^t_{srf,pnd} & \text{and} \\ R^t_{srf,sl} &= \begin{cases} I^t_{pot,sl} - I^t_{cap} & \text{if } I^t_{pot} > I^t_{cap} \\ 0 & \text{otherwise} \end{cases} & \text{and} \\ R^t_{srf,pnd} &= \begin{cases} I^t_{pot,pnd} - I^t_{cap} \cdot fc^t_{pnd} - I^t_{dpr} & \text{if } I^t_{pot} > I^t_{cap} \cdot fc^t_{pnd} + I^t_{dpr} \\ 0 & \text{otherwise} \end{cases} & \text{and} \\ R^t_{srf,pnd} &= \sum_{nsoil} \frac{\theta^t_{upd,i} - \theta^t_{sat,i}}{\Delta t} & \text{for all } \theta^t_{upd,i} > \theta^t_{sat,i}. \end{split}$$


Here,  $R^t_{srf,h}$  constitutes the Horton overland flow, with  $R^t_{srf,sl}$  referring to the runoff that is generated across the entirety of the tile area and  $R^t_{srf,pnd}$  to the additional overflow of the surface depression storage.  $R^t_{srf,d}$  constitutes the Dunne-type overland flow, with  $\theta^t_{upd,i}$  being the soil moisture on layer i after the state of the soil has been updated (with infiltration, as the upper boundary condition, limited by  $I^t_{cap}$ ,  $I^t_{pot,sl}$  and  $I^t_{pot,pnd}$ ) and  $\theta^t_{sat,i}$  the soil porosity.


The (lateral) subsurface runoff  $(R^t_{blg,i})$  generation on a given layer i of a tile t, is determined as a function of the hydraulic conductivity,  $k^t_i$ , and the effective slope  $m^t$  (see sec. 2.2):






$$R_{blg,i}^t = k_i^t \cdot \sin\left(m^t \cdot \frac{pi}{2}\right),\tag{13}$$

Since high-resolution grid-cells, tiles and even the plots in site-level simulations still cover a given area, the use of the above point-scale formulations is not without problems. Most importantly it can not be assumed that the generated runoff reaches any tributary or connected, lower situated area instantaneously. Consequently, the fluxes are initially stored in intermediary reservoirs which, conceptually, represent the flow paths along which the water moves downslope. With respect to the overland flow we assume that the fluxes are comparatively fast allowing us to neglect any interactions with the underlying ground or the overlying atmosphere. Thus, the outflow  $(R_{srf}^{t,*})$ , that is the rate at which water reaches the connected downslope areas or the stream system, is exclusively a function of the water content of the intermediary reservoir  $(L_{srf}^t)$  and the assumed retention time  $(\tau_{srf}^t)$ ; see below):

$$R_{srf}^{t,*} = \frac{L_{srf}^t}{\tau_{srf}^t}. (14)$$

The outflow  $(R_{blg,i}^{t,*})$  from a given layer (i) of the below-ground reservoir  $(L_{blg,i}^t)$  is determined analogously:

$$R_{blg,i}^{t,*} = \frac{L_{blg,i}^t}{\tau_{blq}^t}.$$
 (15)

Here, however, interactions with the surrounding soil can not be neglected. On the one hand, the lateral movement is comparatively slow, allowing the water to percolate downwards and drain beyond the bedrock boundary. On the other hand, the lateral movement is not independent of the level of saturation of the soil, even though a large fraction of the flow may occur through preferential flow networks, partially disconnecting it from the soil matrix. To account for these two effects, the water content of the below-ground reservoir is reduced by a bottom drainage flux  $(R^t_{bot,i})$  and by a (soil moisture) restoration flow  $(R^t_{res,i})$  which moves water from the intermediary reservoir back into the pore spaces, once the water content of a given layer drops below a certain threshold. For the bottom drainage flux we assume that the water drains freely from the reservoir, with the drainage rate being limited to the saturation hydraulic conductivity of (fractured) bedrock,  $(k_{rock})$ ; assumed to be  $1 \cdot 10^{-7}$  m s<sup>-1</sup>) and the hydraulic conductivity  $(k_b)$  of the soil layer containing the bedrock interface (b). Here, the water is not removed from the soil starting at the bedrock boundary – i.e. the lowest layer of the intermediary reservoir – but from the top of the soil column. This is done to account for the vertical movement within the reservoir, assuming that water percolates downward with the rate it drains into the bedrock. With respect to the restoration flow, we assume that these fluxes occur once the excess water has drained from the soil, setting the moisture threshold to the field capacity  $(\theta^t_{fc,i})$  of a given soil layer.

$$\frac{\Delta L_{blg,i}^{t}}{\Delta t} = R_{blg,i}^{t} - R_{blg,i}^{t,*} - R_{res,i}^{t} - R_{bot,i}^{t}, \qquad \text{with}$$

$$R_{blg,i}^{t,*} = \frac{L_{blg,i}^{t}}{\tau_{blg,i}^{t}}, \qquad \text{and}$$

$$R_{res,i}^{t} = \begin{cases} 0, & \text{if } \theta_{fc,i}^{t} > \theta_{i}^{t} \\ \frac{\theta_{fc,i}^{t} - \theta_{i}^{t}}{\Delta t}, & \text{if } \theta_{fc,i}^{t} < \theta_{i}^{t} \text{and } \theta_{fc,i}^{t} - \theta_{i}^{t} < L_{blg,i}^{t} \\ \frac{L_{blg,i}^{t}}{\Delta t} & \text{otherwise} \end{cases}$$

$$R_{bot,i}^{t} = \begin{cases} 0. & \text{if } \frac{\sum_{l=1}^{i-1} L_{blg,l}^{t}}{\Delta t} > k_{bot} \\ \frac{L_{blg,i}^{t}}{\Delta t} & \text{if } \frac{\sum_{l=1}^{i-1} L_{blg,l}^{t}}{\Delta t} 







Di Prima et al., 2018). The latter may be accounted for in the estimate of the transmissivity via a macropore enhancement of the hydraulic conductivity (Milly et al., 2014). However, given the uncertainty of the factors involved and the simple nature of our approach, we instead use globally uniform values for the flow velocities, assuming a value of  $1 m h^{-1}$  for  $v_{blg}$ , with the order of magnitude corresponding to the conductivity of soils including the effects of macropores and pipes or the saturated hydraulic conductivity of sand. Here, the values are independent of the degree of saturation of the soil, meaning there is no distinction between groundwater- and interflow. This was done because the model strongly inhibits the lateral fluxes in the vadose zone, since the water in the intermediary reservoir migrates back into the soil column of the runoff-generating tile once the latter starts drying and soil moisture levels drop below the field capacity. For the velocity of the overland flow, we simply assume that it is an order of magnitude larger than the below-ground values, setting  $v_{srf}$  to  $10 m h^{-1}$ , which is at the lower end of the range of previous estimates (McCaig, 1983).

An accurate description of the lateral fluxes on the macro-scale is even more difficult as it involves additional uncertainty. On the catchment- or hill-slope-scale, the distances are usually large enough to allow water to percolate downwards to the bedrock boundary, where horizontal fluxes occur as sheet flow or through preferential flow networks, possibly returning to the surface at the bottom of the slope (Kirkby, 1988). Approaches that use Darcy's law to describe the long-distance lateral transport across sloped terrain risk underestimating the subsurface runoff, with observed lateral flow velocities sometimes being orders of magnitude larger than what can be expected from a flow through a porous medium (Graham et al., 2010). Since our model does not include a representation of the respective dynamics, these fluxes need to be parameterized. At least for the surface runoff, it is possible to estimate specific lag factors – analogously to the determination of unit-hydrographs – based on the geomorphological characteristics of the grid-cell, while the velocity of sub-surface runoff may be approximated based on observed rainfall and base flow rates (Lohmann et al., 1996; Kumar et al., 2007; Lehner et al., 2008; Singh et al., 2014; Mizukami et al., 2016). For the present implementation, however, we use a simpler approach and calculate the macroscale lag factors  $f_{mac \ l}^{lag,t,s}$  using the same assumptions as the ARNO-scheme, namely that the drainage of excess water from any catchment-sized area takes place over a period of a few days (Todini, 1996; Dümenil and Todini, 1992). In the scheme's implementation in ICON-Land, the minimum and maximum drainage rates are globally uniform and independent of the model resolution, which further assumes that the overall retention time is largely determined by the time it takes water to pass through the ground and that the distances to the nearest tributary are not only similar in all drainage basins but also small relative to the horizontal grid spacing of the model. Accordingly, we determine the lag factors  $(f_{mac,l}^{lag,t,s})$  based on the time-step length and globally uniform retention times  $(\tau_{mac,l}^{ret})$ :

$$f_{mac,l}^{lag,t,s} = \frac{\Delta t}{\tau_{mac,l}^{ret}}.$$
 (18)

For standard applications, in which the macro-scale transport moves water across typical drainage-basin scales,  $\tau_{mac,blg}^{ret}$  is set to  $120\,h$  and  $\tau_{mac,srf}^{ret}$  is set to  $10\,h$  again assuming that the flow velocities at the surface are a magnitude larger than those below ground. In case that the distances represented by the macro-scale connections are below the catchment scale,  $\tau_{mac,blg}^{ret}$ 


and  $\tau_{mac,srf}^{ret}$  can be specified via a namelist parameter. However, for T-REX applications that implicitly resolve slopes by a number of tiles (e.g. corresponding to different height bands), it may be more appropriate not do so using macro-scale connections. In such cases, the relations between tiles should rather be based on micro-scale connectivities where distances are explicitly accounted for in the estimation of the lag factors.

The weighted sum off the sibling-specific lag factors, constitutes the average outflow lag factors of each tile ( $f_{mic,l}^{lag,t}$ ) and  $f_{mac,l}^{lag,t}$ ). For macro-scale connections, the additional assumption is being made that for those fractions of the tile that feature micro-scale connections, macro-scale fluxes may not occur. Here, the average lag factors are scaled according to the cover fraction of the tile and the cover fractions of the (micro-) connected sibling-tiles:

$$f_{mic,l}^{lag,t} = \sum_{s=1}^{nct} f_{mic,l}^{lag,t,s} \cdot c_{hfp*,mic,l}^{t,s} \qquad \text{and} \qquad (19)$$

$$f_{mac,l}^{lag,t} = \sum_{s=1}^{nct} f_{mac,l}^{lag,t,s} \cdot c_{hfp*,mac,l}^{t,s} \qquad \text{with}$$

$$c_{hfp*,mic,l}^{t,s} = \frac{fc^s \cdot c_{hfp,mic,l}^{t,s}}{\sum_{k=1}^{nct} fc^k \cdot c_{hfp,mic,l}^{t,k}} \qquad \text{and}$$

$$c_{hfp*,mac,l}^{t,s} = \begin{cases} \frac{fc^s \cdot c_{hfp,mac,l}^{t,s}}{\sum_{k=1}^{nct} fc^k \cdot c_{hfp,mac,l}^{t,k}} \cdot \frac{fc^t - \sum_{k=1}^{nct} c_{hfp,mic,l}^{t,k} \cdot fc^k}{fc^t} & \text{if } fc^t - \sum_{k=1}^{nct} c_{hfp,mic,l}^{t,k} \cdot fc^k > 0. \\ 0. & \text{otherwise.} \end{cases}$$

The average retention times for each tile  $(\tau_l^t$ , where l either stands for surface [srf] or below-ground [blg]), are calculated based on the average lag factors for surface and below-ground fluxes  $(f_l^{lag,t})$ , which can be obtained simply by adding the respective lag factors for fluxes on the micro- and macro scale:

$$\tau_l^t = \frac{\Delta t}{f_l^{lag,t}} \qquad \text{with}$$
 
$$f_l^{lag,t} = f_{mic,l}^{lag,t} + f_{mac,l}^{lag,t}. \tag{20}$$

Finally, the fraction of the surface- and subsurface runoff  $(p_l^{t,s})$  from tile (t) received by any of its siblings (s) is determined by the ratio of the sibling-specific lag factor and the average lag factor. For tiles that are not fully connected to downstream tiles - i.e.  $\sum_{k=1}^{nct} c_{hfp,mic,l}^{t,k} + c_{hfp,mac,l}^{t,k} 

Applying this partitioning to the fluxes from the intermediary reservoirs  $(R_l^{t,*})$ , the lateral runoff flux  $(R_l^{t,s,*})$  from tile t to any of its siblings s is calculated according to:

$$R_l^{t,s,*} = R_l^{t,*} \cdot p_l^{t,s}. \tag{22}$$

In case of overland flow, the runoff contributes to the infiltratable water of the receiving sibling tile ( $I_{pot,sl}^s$  and  $I_{pot,pnd}^s$ ), with the inflow primarily increasing the water content of the surface depression storage ( $\Delta W_{sfc}^s$ ):

$$I_{pot,sl}^{s} = P \cdot (1 - fc_{pnd,mx}^{s}) + R_{srf}^{t,s,*} - \frac{\Delta W_{sfc}^{s}}{\Delta t}$$
 and (23) 
$$I_{pot,pnd}^{s} = P \cdot fc_{pnd,mx}^{s}) + \frac{W_{sfc}^{s} + \Delta W_{sfc}^{s}}{\Delta t}$$
 with 
$$\Delta W_{sfc}^{s} = \begin{cases} R_{srf}^{t,s,*} \cdot \Delta t & \text{if } R_{srf}^{t,s,*} \cdot \Delta t 


$$\Delta D_{snw}^{h,l} = \frac{c_{ctc*}^{h,l}}{\sum_{k=1}^{nct} c_{ctc*}^{h,k} + c_{ctc*}^{k,h}} \cdot \begin{cases} D_{snw} \left( \frac{1}{f_{c,rel}^{l}} - 1 \right) & \text{if } D_{snw} 

Similarly, standing water at the surface does not have an explicit temperature, with the exception being large lakes, but these are represented by separate tiles that do not interact with the surrounding land surface. Furthermore, the heat storage of (all other) surface water bodies, is not accounted for in the surface energy balance of the model. This makes it impossible to simulate the convective heat fluxes due to surface runoff, with the latter initially being given to the surface water reservoir of the receiving tile before it infiltrates into the ground or runs off. Here an optional formulation was introduced into the model which modifies the surface heat capacity and is based on the assumptions that, i) any precipitation has the same temperature as the surface (which is the same assumption as in the standard model), ii) that the surface water bodies have no thermal stratification – which is a permissible simplification as long as the surface water bodies are limited to puddles, ponds and politic lakes – and iii) that the heat exchange between surface water and the ground is sufficiently fast for the uppermost soil layer to have approximately the same temperature as the overlying surface water body. This modified volumetric heat capacity of the top soil layer is calculated as:

$$c_{soil,1}^* = \frac{c_{soil,1} \cdot \Delta z_1 + c_w \cdot h_{wtr} + c_{ice} \cdot h_{ice}}{\Delta z_1}$$
(28)

Using the modified heat capacity, any change in temperature due to a surface convective heat flux calculated by eq. 27 (with  $\Delta T_{i*}^{t,s}$  referring to the uppermost soil layer of both tiles and  $W_{i*}^{t,s}$  being the surface runoff) is (implicitly) being applied to the surface water body along with the uppermost soil layer.

With large time steps and lateral fluxes of water, it is possible that the lateral heat fluxes calculated above could increase (lower) the temperature on the receiving tile above (below) those of the tile the water originated on. To prevent this, the lateral heat fluxes are limited by the temperature differences and the heat capacity of the receiving tile:

$$C_{i*}^{t,s} = \Delta T_{i*}^{t,s} \cdot \frac{c_{soil,i*}^s \cdot \Delta z_{i*}}{\Delta t} \qquad \text{if } |C_{i*}^{t,s}| > |\Delta T_{i*}^{t,s} \cdot \frac{c_{soil,i*}^s \cdot \Delta z_{i*}}{\Delta t}|. \tag{29}$$

## 480 3 Example applications



# 3.1 Lateral water transport

For the first example application, we aim to investigate the effect of micro- and macro-scale lateral transport processes on the land's capacity to retain water and the resulting impacts on the turbulent land-atmosphere exchange and surface temperatures. On the macro scale, we separate the grid cell into those parts that generate runoff, the uplands, and those that receive the generated runoff, i.e. the lowlands. With respect to the micro scale, the setup distinguishes between those grid-cell areas where water may pool on top of the surface, i.e. local depressions, and those where excess water runs of immediately, i.e. relative elevations. The respective fractions are determined by combining the 30 m resolution version of the Copernicus DEM (European Space Agency, 2024) together with the CEH CTI dataset (Marthews et al., 2015a) at 15" resolution as follows: In a first step,








we use the Whitebox Tools Open Core (Lindsay, 2014) to fill all local depressions in the DEM data. The difference between this and the original data provides us with the depression depth. Next, we aggregate this data to a 15" resolution thereby also computing the depression fraction as grid-cell area belonging to local depressions relative to the total cell area. Furthermore, we use the Connected-Components implementation of the OpenCV Toolbox (Bradski, 2000) to label all grid cells belonging to the same depression. Thus, we can divide the overall depression area by the number of different depressions to gain an estimate of the mean depression size for each 15" cell. These depression characteristics are then cross-referenced with the CTI data. Wherever the  $CTI \le 5.7$  we consider the cell to be an upland and therefore assign the depression fraction, depth and average size to the upland region and vice versa for lowlands with a CTI> 5.7. Here, we chose a threshold value of 5.7, because it is the median of the global dataset and because it roughly corresponds to the value commonly used separate lowlands from uplands and slopes (Marthews et al., 2015b). Finally, the data is further aggregated onto the R2B4 grid ( $\approx 160$  km resolution) which is used for our global simulations. Grid cells with missing values are set to the average of their respective quantities. We conduct ICON-Land-standalone simulations driven by climate forcing from the Global Soil Wetness Project Phase 3 (GSWP3, Dirmeyer et al. (2006); Kim (2017)). The simulations were performed for the period 1979-2018, with the analysis using the model output from the period 2009-2018. In the following, we compare simulations in which all subgrid-scale lateral transport of water is disabled to simulation in which either micro-scale or macro-scale transport processes are accounted for as well as a simulation in which transport processes on both scales are enabled in the model.

Water is transported laterally mainly in those regions of the world where the water availability at the land surface exceeds the atmospheric moisture demand, at least temporarily. Thus, the model simulates only small lateral subgrid-scale fluxes in the arid and semi-arid regions, while they can exceed 500 mm yr<sup>-1</sup> in the humid areas (Fig. 5 a). However, also the humid regions in the tropics feature extensive areas with comparatively little lateral water movement, such as the Amazon basin. Here, it is most often the topography that does not favor large quantities of water being moved laterally through the soil or at the surface. Comparatively small slopes in the contributing areas generate only little surface- and lateral subsurface runoff and, in the model, most of the water drains at the bottom of the soil column, contributing to the channel flow as base-flow below the bedrock boundary (not shown). In contrast, the flat areas in the cold regions often feature larger lateral fluxes, since here the ground is predominantly frozen, at least during the snowmelt season, and water can not percolate deep into the soil but is forced to run off laterally within a comparatively shallow layer close to the surface.

Lateral transport processes enable low-lying parts of the gridcell to store the runoff from upslope areas and allow a redistribution of water between those areas with a large depression storage and those that may not retain water at the surface. Thus, they almost exclusively increases the terrestrial water storage (Fig. 5 d), with marked increases in the total soil water content and in the wetland area (not shown). There are, however, a few grid cells, mainly in the (sub-) polar regions, where the total terrestrial water, in particular the soil water content, is reduced. Here, the lateral transport initially increases the moisture content in the near-surface layers after snowmelt, which increases the heat conductivity of the ground (not shown). This, in turn, raises the ground heat flux and allows a higher fraction of the soil ice to melt. And with more water being liquid, the resulting increase



Figure 5. Effects of the lateral water transport on terrestrial water storage, latent heat flux and sugrid-scale temperature variability

Shown is the total subgrig-scale lateral flux as the sum of, surface runoff, lateral drainage and diffusive fluxes for a simulation in which microand macro-scale transport processes are enabled (a), a simulation in which only the micro-scale transport is active (b) and a simulation that only accounts for the macro-scale lateral fluxes (c). d,e,f, show the resulting impacts on the total terrestrial water storage. These were calculated as the differences between a given simulation with (parts of) the lateral transport activated and a reference simulation in which the lateral transport is switched off. g,h,i show the impact on the grid-cell mean latent heat flux, while j,k,l show the change in the maximum spatial (subgrid-scale) temperature variability. The later was calculated as the temperature difference between the warmest and coldest tile within a given grid cell.

in evapotranspiration and drainage during summer and fall causes an overall reduction in the annual mean soil water content. Substantial increases in the terrestrial water storage are mainly limited to the temperate zone and the (sub) polar region, where the additional retention capacity allows to partially compensate the differences in the seasonality of water availability and (atmospheric) moisture demand. Depressions, in particular, can store parts of the spring snowmelt, with the water subsequently diffusing into the surrounding areas increasing the water availability during spring and summer when potential evaporation rates are high. Here, the effect on the simulated wetland extent is especially pronounced and increases in the inundated area – relative to the potentially inundated area – of up to 70 % across large parts of the temperate and most of the polar and subpolar regions show that lateral exchange processes are essential for maintaining standing water at the surface (not shown).





Pronounced effects on the total soil water storage are, however, limited to those regions that feature a deep bedrock boundary, where increases in total soil water of up to 0.2 m often correspond to a relative increase in the soil water content of about 20 % (not shown). In regions where the bedrock boundary is comparatively shallow, such as most of eastern Siberia, there are only small increases in the soil water content, despite large lateral fluxes. In contrast, some large effects are observable in regions where the lateral fluxes are comparatively small, such as the West Siberian Lowlands, where a large depression capacity, deep soils and low potential evaporation rates facilitate the water storage over longer periods allowing even comparatively small lateral fluxes to have a notable impact on the soil moisture and depression storage.

Within the subtropics and the tropics, the effects on the terrestrial water storage are largely negligible (Fig. 5 d). Here, most precipitation transpires and evaporates locally and the model simulates only minor lateral fluxes. Additionally, the temporal offset between water availability and demand is much smaller than in the temperate and polar region and most of the water that is redistributed laterally evaporates or is transpired swiftly rather than being stored in the soil or in depressions over longer periods. Consequently, there are areas in the tropics and subtropics with only a minor impact on the terrestrial water storage but with a notable effect on the latent heat flux (Fig. 5 g). In general, the inclusion of the lateral exchange fluxes increases the 545 latent heat flux notably, mainly in those areas that also feature large changes in the terrestrial water storage where respective effects can correspond to relative increases of up to 10 %. And since the increase in latent heat flux is mainly balanced by a decrease in the sensible heat flux, the Bowen ratio changes by as much as 20 %. Yet even a 20-percent change in the Bowen ratio does not drastically alter the overall surface energy balance on the grid-cell level. Consequently, the effects on the (grid cell) average surface temperatures are comparatively small and the cooling due to the inclusion of the lateral water transport rarely 550 exceeds -0.2 K (not shown). However, the spatial subgrid-scale temperature variability is extremely sensitive to the inclusion of the lateral transport processes, since the local effects on the Bowen ratio in the lowlands and depressions can far exceed the grid-cell mean value. Here, the temperature differences between the warmest and the coldest grid-cell fraction can increase by as much as 3 K (Fig. 5 j), corresponding to an order of magnitude increase across most of North America and Eastern Siberia.

A comparison of the micro- (Fig. 5 b) and the macro-scale fluxes (Fig. 5 c) reveals a large similarity with respect to the spatial patterns and there are only few areas in which the model simulates relatively large micro-scale fluxes but only relatively small macro-scale fluxes (and vice versa). This suggests that the global patterns are mainly controlled by scale-independent factors, such as the state of the atmosphere – which determines precipitation and potential evaporation but also the general thermal state of the soil (frozen or unfrozen) – and the soil properties, and less by the specific topography within a given grid cell. In contrast the overall magnitude is quite different, with the lateral fluxes being predominantly larger for the micro-scale exchange – on average by about 40 %. With respect to the total terrestrial water storage (Fig. 5 e, f), the effect of the micro-scale transport is about 25 % larger than that of the macro-scale transport, with the impact on the total inundated area being almost 50 % larger. In contrast, the increase in latent heat flux is actually about 15 % smaller for the micro- than for the macro-scale lateral fluxes, due to the different ways the water is redistributed within the grid-cell system (Fig. 5 h, i). Here, macro-scale runoff increases the soil moisture in a comparatively large area, i.e. the lowland region, while the micro-scale transport pro-








cesses predominantly lead to a moisture convergence in a comparatively small part of the grid cell, i.e. the depressions. And since evapotranspiration rates have a non-linear dependency on the water availability, most importantly they are limited by the available energy, a small increase in the soil moisture within a large region has a stronger effect on the latent heat flux than a large increase in soil moisture in a small region. The above, however, also means, that the effect on the maximum (spatial) temperature gradient is notably larger – on average by about 35% – for the micro-scale fluxes (Fig. 5 k, 1). Due to the moisture convergence in a comparatively small part of the grid cell, the local temperatures within the depressions predominantly show a stronger reduction due to micro-scale fluxes, than the temperatures in the lowland fraction due to the macro-scale fluxes. Finally, it should be noted that the effects in the simulation that enable both micro- and macro-scale transport processes are not necessarily the same as combining the effects from the micro-scale-flux-enabling and the macro-scale-flux-enabling simulations. The lateral-movement of water in the fully-enabled simulation is very close to the sum of the effects in the two simulations that only enable one flux component (Fig. 5 a, b, c). However, in case of the terrestrial water storage the former is about 25 % smaller than the sum of the effects (Fig. 5 d, e, f) and for the effect on the latent heat flux it only amounts to about 60 % (Fig. 5 g, h, i). This discrepancy arises because the terrestrial water storage and the latent heat flux are not only limited by the available water, but additionally by the pore volume and the potential depression storage – in case of the terrestrial water storage – and by the available energy – in case of the latent heat fluxes. Thus, an increase in the laterally transported water does not necessarily entail an increase in the water storage or in evapotranspiration.

## 3.2 Lateral heat transport

In the second application, we aim to investigate the effect of the lateral heat transport on the subgrid-scale temperature variability. Here, we set up the model to represent typical patterned-ground structures – e.g. non-sorted circles, ice-wedge polygons – that are often found in the arctic tundra. The presence of such patterns is limited to periglacial regions and, while the coverage of environmentally suitable spaces has been estimated at around 3 Mio. km<sup>2</sup> for ice-wedge polygons (Karjalainen et al., 2020), other studies conclude the actual extent of the polygonal tundra to be merely around 0.3 Mio. km<sup>2</sup> (Höfle et al., 2013). However, these structures present an ideal test case for our model, since soil organic matter concentrations, and with that hydrological and thermal soil properties, can vary dramatically on the sub-meter scale (Ping et al., 2015). For this application, we assume circular structures, that consists of an organic-rich center, with a volumetric organic matter fraction of 95 % at the surface – i.e. the uppermost 0.1 m of the soil, encompassed by the rim section, which has a near-surface soil organic matter fraction of 35 %. These two are in turn surrounded by an outer section with mainly mineral soil throughout the below ground column, i.e. an organic matter fraction of 5 % at the surface. We assume the radius of these circles to be evenly subdivided into the center, rim and outer section, with water draining from the circle either at the soil bedrock interface or laterally through the outer section, with the former only being possible if the active layer is sufficiently deep. For simplicity reasons, we assume these circles to be omnipresent in the permafrost region, allowing us to conduct our analysis for the pan-Arctic average rather than focusing on a specific grid cell. We conduct ICON-Land-standalone simulations driven by climate forcing from the Global Soil Wetness Project Phase 3 (GSWP3, Dirmeyer et al. (2006); Kim (2017)). The simulations were performed for the







period 1979-2018, with the analysis using the model output from the period 2009-2018. For different setups we compare the spatial, below-ground temperature variability, i.e. the temperature differences between center, rim and outer section, between simulations in which the lateral heat transport is disabled and simulations with the heat transport active.

In a first step, we assume a radius of one meter, with center, rim and outer section each covering 0.33 m. Additionally, we assume the circle to be level so that there is no vertical offset between the surface of the sections. The latter may not be the most common configuration, since the circular structures often have a distinct micro-topography, e.g. an elevated center in case of a hummock-type structure, but it makes the analysis much more straight-forward, since effects due to the lateral water transport and a lateral coupling with a vertical surface offset are (largely) absent. More importantly, it allows us to compare our simulations with those of the Dynamic Soil Model (Thurner et al., 2025, DynSoM), a pedon-scale soil model, which was developed to explore the movement of energy specifically in permafrost-affected soils. DynSoM simulates the key physical variables, such as soil temperature, and soil water and ice content, as well as the surface and below-ground heat fluxes both in the vertical and horizontal direction (note that the lateral movement of water is not represented in the present DynSom setup). Here, DynSoM, in contrast to ICON-Land, is a true 2D model which explicitly resolves the soil in one horizontal direction. For the present setup, the model uses a 10-centimeter horizontal grid-spacing and a variable spacing in the vertical direction, with the uppermost meter of the soil column being resolved in 10-centimeter steps. The simulations with DynSoM are based a 1-m soil transect of a non-sorted circle in the Chersky region (Gentsch et al. (2015); Supplements profile CH-E [note that in the original transect the inner circle – i.e. the circle center – features the low organic matter concentration and the inter circle area – i.e. the outer section – the high concentrations, but the order can simply be reversed without affecting the results of the DynSoM simulation]) and are forced with CRUNCEP data (Viovy, 2018) for the respective region. The setup of the DynSoM soil transect differs to the synthetic ICON-Land setup in that the organic rich region (surface organic matter fraction of 95 %) and the organic poor region (organic matter fraction of 5 %) both only cover 0.2 m, while the intermediary region (surface organic matter fraction of 35 %) covers 0.6 m. Furthermore, the mineral soil below the top layer also shows differences along the transect, while the present ICON-Land setup assumes spatially homogeneous mineral soil properties within one grid cell. However, since we do not attempt a site-level validation of our model but merely aim to compare its general behavior to a model that explicitly resolves the horizontal exchange on the pedon-scale, we assume that respective setups of the models are sufficiently similar.

In DynSoM, soil organic matter predominantly lowers the soil heat conductivity, which impedes the ground heat fluxes during the snow-free season (not shown). When neglecting the lateral exchange, this entails notably lower soil temperatures in the organic rich center than in the rim and the outer section, with differences at the surface starting to emerge after the snowmelt in early summer (Fig. 6; a, b). The signal subsequently propagates downwards and the peak temperature differences in depth > 2 m are being reached the following spring. At the same time, the lower heat conductivity in the center also results in a more pronounced heating of the surface during late summer and a less pronounced cooling during early fall, with the near-surface temperatures exceeding those in the rim and the outer section between August and October. However, as soon as the soil is

Figure 6. Effects of the lateral heat transport on spatial temperature differences in patterned ground

Shown are the monthly temperature differences until a depth of 5 m between the organic rich circle center and the rim (first and third column) and between the center and the outer section (second and fourth column), for simulations that do not account for lateral transport processes (first and second column) and simulations where these processes are included in the model (third and fourth column). The first row shows the differences between simulations with the DynSoM model, for a flat-centered circle with a total radius of 1 m. The second row shows the corresponding ICON-Land simulations. The third row shows the temperature differences in a ICON-Land simulations for a circle radius of 10 m and the fourth row for a circle radius of 100 m. The fifth row shows the temperature differences for a ICON-Land setup of a low-centered circle with a 1-m radius and the last row for a setup of a high-centered circle with a 1-m radius.

insulted by an increasingly thick snow cover in fall, also the near surface temperatures in the center fall below those of the rim and the outer section. During the cold season, the lower heat conductivity of the organic matter plays a subordinate role








mainly because the soil is insulted by the snow cover. Additionally, ice has a much higher heat conductivity than liquid water and the relative differences in the near-surface heat conductivity between the sections are much smaller when the soils are frozen. Consequently, the cooling of the center, relative to the rim and the outer section, during summer is not fully balanced by a relative warming during the cold season. A temperature convergence during the cold season is observable in some soil layers, however, the temperatures in the center remain below those in the rim and the outer section even in these layers. Here, the temperature differences between center and rim and between center and outer section are largely similar. However, the signal is slightly more pronounced for the center-rim comparison, despite the near-surface soil organic matter concentrations in the center actually being closer to those in the rim than those in the outer section. The reason for this lays in the fact that, at depth > 5 m, the rim has a higher heat conductivity than the outer section, which has a notable effect on the magnitude of the bottom heat flux (not shown). Here, DySoM doesn't employ a zero-flux condition, but a prescribed temperature. With 0°C, this temperature is higher than the simulated soil temperatures in the superjacent layers resulting in a positive bottom heat flux. Owing to the higher conductivities, the rim exhibits a larger warming at the bottom of the soil column, hence, higher temperatures than the outer section. Thus, the temperature differences to the cooler center are also more pronounced for the rim.

In the ICON-Land simulation, the temperature differences between the sections are equally driven by differences in the toplayer heat conductivity during late spring and summer (Fig. 6; e, f). Consequently, the overall effects are quite similar between the two models, especially given the fact that we are comparing simulations of a specific site (DynSoM) to the pan-Arctic average based on a synthetic setup (ICON-Land). However, there are also important differences, most notably in that large temperature differences between the sections are confined to the uppermost meter of the column and that the near-surface temperatures during the cold season are actually higher in the circle center. Both of these differences are related to the hydraulic properties of the soil organic matter – which can differ quite strongly to those of the mineral soil fractions – and to the assumed drainage pathways of the circle. Contrary to the DySoM simulation, the soil in the ICON-Land simulation is not necessarily water saturated and the degree of saturation of the soil in a given circle section depends on the overall water holding capacity, the infiltration rates and the position on the drainage path. Here, organic matter has a higher porosity than the mineral soil fractions – hence can hold more water – and also a higher hydraulic conductivity – hence allows for higher infiltration rates. Furthermore, the circle center drains laterally through the rim and the outer section and without vertical offset between the three sections, lateral fluxes mainly occur if the rim and the outer section are less saturated than the circle center. As a result, the soil column is more water saturated in the organic rich center than in the rim and the outer section, which entails higher heat conductivities below the organic layer (not shown). During the snow-free season, when the soil warms from the top, the combination of lower heat conductivities at the surface and higher heat conductivities below does not only reduce the overall heat uptake, but also enhances the downward transport, resulting in a reduced vertical temperature gradient. This difference in heat distribution causes the cooling of the center (relative to the rim and the outer section) to be more pronounced closer to the surface and less pronounced deeper within the soil. Similarly, the higher heat conductivities also lead to a reduced vertical temperature gradient during the cold season. However, during this period, the column cools from the top and the reduced temperature gradient corresponds to an enhanced upwards heat transport in the center and, consequently, higher near-surface






temperatures than in the rim and the outer section. Here, the respective effects are more pronounced in the center-outer-section than in the center-rim comparison, i.e. larger positive and negative temperature differences close to the surface and a slightly less pronounced relative cooling below depths > 1m. This is because the near-surface soil organic matter concentrations in the center are closer to those in the rim than those in the outer section and because the soil column in the latter, as the outermost circle section, is predominantly less water saturated than in the rim.

As discussed above, there are some differences in the general effect of a heterogeneous soil organic matter distribution between the models. Nonetheless, they agree well on the impact of the lateral heat fluxes on the spatial temperature variability. In both simulations, the temperature differences between the circle sections are strongly reduced and any notable spatial variability is largely limited to the uppermost meter of the column as well as to the snow-free season (Fig.6; c, d, g,h). Here, the fact that the lateral heat fluxes have a stronger influence on the temperature differences at greater depths has two main causes. On the one hand, the temperature differences deeper within the soil are not only balanced by the lateral heat fluxes on the respective layers, but they are also reduced since the vertical heat exchange with the superjacent layers is spatially more homogeneous due to the lateral heat exchange in the layers above. On the other hand, the temporal temperature variability is strongly dampened with increasing depth. Thus, while large near-surface temperature differences between the sections may not be fully balanced on the short term, the signal is attenuated while propagated downwards to the extent that temperature differences in the deeper soil layers only emerge if the longer-term mean temperature near the surface is notably different.

Overall, the above results suggest that, at the pedon-scale, the lateral heat fluxes are sufficiently strong to largely balance the below-ground temperature differences resulting from the spatial variability in the surface heat fluxes. Furthermore, it appears that these effects can be captured reasonably well by LSMs employing a tiling-approach. The latter allows us to use ICON-Land simulations to investigate the extent to which the ability of the lateral heat exchange to the equilibrate soil temperatures depends on the horizontal distances considered. For a radius of 10 m, with the circle sections each covering 3.33 m, we find that the temperature differences between center, rim and outer section above 1 m, appear to be hardly affected by the lateral heat exchange (Fig.6; i, j, k,l). This strongly suggest that at these length scales the lateral heat exchange is too weak to balance the short-term effects resulting from the differences in the seasonal variability of the surface heat fluxes. However, the near-surface temperatures in the center do in fact show a slight longer-term increase due to the lateral fluxes (e.g. for the center-outer-section comparison this is visible in the [positive] temperature differences during the cold season being slightly larger) and the resulting annual mean temperature is very close to that in the rim and the outer section. Consequently, in those soil layers that are less sensitive to the seasonal variability in the surface heat flux, the spatial temperature variability is reduced substantially. At depths > 1 m, there are no marked temperature differences neither between the center and rim nor between center and the outer section. Here, it should be noted that the absence of large temperature differences comparatively close to the surface is partly caused by the rather coarse vertical resolution of the model, since the damping of the temporal temperature variability depends on the latter. With a thickness of almost 3 m, the first soil layer below a depth of 1 m possesses a substantial thermal inertia and the temporal variability of the vertical heat fluxes only has a small effect on the temperatures of this layer. Thus it



is highly plausible that, in reality or with a higher vertical resolution, the spatial temperature variability extends much further downward than in our simulation. Increasing the radius to 100 m, with the sections each covering 33.33 m, further reduces the lateral heat fluxes. As a result, the latter are not only too weak to balance the short-term effects due to the variability of the surface heat fluxes, but also the longer-term mean state of the near surface layers is substantially different. Consequently, notable differences in the below-ground temperatures persist even at depths > 4 m and, for these length scales, the impact of the lateral heat fluxes appears to become negligible (Fig.6; m, n, o,p).

As stated above, a level surface of the circle may not be the most common configuration as in reality we often find highcentered hummock-type structures, or low-centered hollow-type structures. On the one hand, the vertical offset between the sections of the circle affects the soil thermophysical properties, since it determines the flow of water, hence, the saturation of the soil and, thusly, the heat conductivity. On the other hand, the ground features a strong vertical temperature gradient during most of the year, with the vertical offset between two sections determining the distance to the surface of any two interfacing points (Fig. Fig.4). Thus, in the final example we investigate the extend to which the micro-scale topography of the circle effects the spatial temperature variability. First we focus on a low-centered circle with a 1-m radius, in which the surface of the center and outer section are both located 0.1 m below the surface of the rim. For, this configuration we find notable differences between the temperatures of the sections despite the small horizontal extent of the circle (Fig.6; q, r, s,t). The near surface soil layers in the center are cooler during summer and warmer during winter, with the differences being larger in the center-rim than in the center-outer-section comparison. Here, the temperature differences between center and rim are less determined by the differences in heat conductivities and mainly reflect the vertical temperature gradient across the uppermost 0.1 m of the soil. The lateral heat fluxes are driven by the temperature differences of two points with the same absolute vertical position, hence the surface temperatures of the center are essentially adjusting to the temperatures at 0.1 m depth in the rim. During the summer the (downwards negative) vertical temperature gradient near the surface reaches a maximum of about 20 K m<sup>-1</sup> and during winter roughly half that value (downwards positive), meaning that temperature differences of up to +2 K and -1 K, are fully explainable by the vertical offset between the two tiles. Furthermore, since the vertical temperature gradient averaged over longer-periods is close to zero, there are no notable differences in the temperatures in the deeper soil layers. The temperature differences between center and the outer section (which have the same vertical offset to the rim) show, however, that there is also a notable variability in the temperatures of points with the same absolute vertical position, i.e. that the surface temperatures in the rim and the outer section do not fully adjust to the temperatures at 0.1 m depth in the rim. This variability is much larger for the low-centered than for the level circle (Fig. 6; h, t), indicating that the lateral heat fluxes are smaller in case of the former. The reason for the reduced lateral exchange lies in the different levels of saturation of the rim. Due to its raised position – in case of the low-centered circle – the rim drains more readily into the other sections, resulting in overall drying of the soil column. This in turn leads to a lower heat conductivity and, consequently, reduced lateral fluxes. For a high-centered structure, in which the surface of the circle center is located 0.1 m above the surface of the rim, which in turn is located 0.1 m above the surface of the outer section, we find that the lateral heat fluxes have essentially the same effect as for a low-centered circle. However, since the center is elevated relative to the rim and the outer section, it features higher near surface temperatures

during summer and colder temperatures during winter (6; u, v, w, x), with the magnitude of the effects increasing with the vertical offset. As a result, the temperature differences, resulting from the lateral heat fluxes actually oppose the ones in the simulation in which the lateral fluxes are being neglected.

## 4 Summary and Discussion

In the present work, we described a new scheme that provides a lateral coupling between the tiles capturing the spatial subgrid-scale heterogeneity of the land surface in ICON-Land, the land component of the ICON framework. The scheme represents 5 lateral exchange processes, namely gravity-driven moisture fluxes and the corresponding convective heat transport, diffusive-and conductive fluxes of water and heat as well as the wind-driven redistribution of snow. The main assumption the scheme is based on is that the relationships between any two homogeneous surface clusters, the tiles, can be described by a set of connectivities. The latter result from the internal logic underlying the tile-definition and reflect the relative (geometrical) contact lengths between two tiles and the dominant hydrological flow paths connecting the two. Taking into account the composition of a given grid cell, they are used to calculate the spatial relationships between two tiles and the time-lag factors that govern the lateral transport. Here, the actual parametrizations of the scheme were chosen to integrate well with the model's existing description of the vertical transport processes, e.g. the scheme merely transports the runoff that is calculated within ICON-Land's point-scale hydrology module, while the time-lag factors for the macro-scale fluxes of water are based on the assumptions underlying the ARNO-rainfall—runoff model, which is used to describe the runoff from catchment-sized areas in standard ICON-Land-setups.



In it's present implementation, the scheme does not provide a final suit of parametrizations that necessarily work for all types of land-surface heterogeneity. And, given the vast range of land processes and their dependencies on different soil and surface properties – all with potentially different spatial distributions, it is highly questionable whether a setup accounting for the entirety of land-surface heterogeneity can even be run with reasonable computational costs. Instead it appears more likely that the model setup, hence, the required lateral-flux parametrizations, will depend on the scientific question that a specific simulation is used to help answer. Thus, our goal was to provide a framework that allows for a consistent treatment of the inter-tile exchange, with enough flexibility to facilitate the implementation of improved or adjusted parametrizations and new processes. Nonetheless, we hope that the two example applications demonstrate the potential usefulness of our scheme, despite its incompleteness. Here, our simulations showed that lateral-transport processes are a key factor determining the subgrid-scale temperature and moisture variability. Among other things they showed that the model's ability to simulate standing water at the surface, was mainly determined by the inclusion of the lateral water movement, with the subrid-scale temperature variability increasing by an order of magnitude across most of North America and Eastern Siberia. Furthermore, the small-scale (spatial) variability in soil temperatures was highly sensitive to the lateral heat fluxes, with the magnitude and even the direction of effects depending on the assumed spatial-scales and the topography.




It should be noted, however, that in our simulations the grid-cell mean state and exchange with the atmosphere did not show substantial changes due to the inclusion of the lateral transport processes. Even more fundamentally, there was very little to suggest that capturing the general physical land-atmosphere interactions necessarily requires a tiling of the land surface and even the wetland extent can most likely be represented sufficiently well with an appropriate parametrization of the depression storage and the retention due to the flow along slopes. In large parts this may be attributable to the nature of the experiments we conducted, since we ran the simulations with prescribed atmospheric conditions, excluding all land-atmosphere feedback effects. Here, fully coupled simulations with a similar setup showed that even comparatively small changes in Bowen ratio can have substantial climate effects in regions where the state of the atmosphere, in particular the low-altitude cloud cover, is sensitive to changes in terrestrial evapotranspiration (de Vrese et al., 2024). Furthermore, a tiling may still be required to represent those processes that have a highly non-linear dependency on the state of the (sub) surface and which may not be well be described based on the grid-cell mean state. Here, one of the most prominent example are the terrestrial methane fluxes, which are dominated by the emissions from soils and wetlands. The latter are the result of anaerobic decomposition processes and, therefore, require anoxic conditions. Since fully water saturated soils can mostly only be found in a fraction of any gridcell-sized area, a tiling approach may still be the most valid strategy to represent the state and fluxes in the respective areas as it allows for a consistent treatment of the local hydrological, thermophysical and biochemical processes. And if tiles are being used to represent different surface clusters, the respective lateral coupling may not be ignored. In case of methane emissions, for example, the lateral exchange fluxes appear to not only have a substantial impact on the extent of areas with fully water saturated soils but also on the subgrid-scale temperature distribution, another key determinant of the local decomposition rates.

Code and data availability. The primary data is subject to the terms of the Creative Commons Attribution 4.0 International license (CC BY
 4.0) while the model code can be used under a permissive open source license (BSD-3C). Both model output and -code are accessible via Zenodo at https://doi.org/10.5281/zenodo.17085112 (de Vrese, 2025).

## Appendix A: Soil hydrology in ICON-Land

ICON-Land includes two approaches for determining infiltration, surface- and subsurface runoff, the first of which is suitable for coarse-resolution simulations and is based on the ARNO-rainfall—runoff model (Dümenil and Todini, 1992; Todini, 1996; Reick et al., 2021). The ARNO model determines the partitioning of the moisture fluxes at the surface based on an assumed subgrid-scale soil moisture distribution. Conceptually, the soil moisture in any grid cell is subdivided into a number of local water storages, the water content (w) of which is assumed to follow the cumulated distribution function:

$$f_{(w)} = 1 - (1 - \frac{w}{w_{max}})^b, \tag{A1}$$

where  $w_{max}$  is the maximum water holding capacity of the local storages and b a steepness-parameter, which accounts for the subgrid-scale topography and is estimated based on the standard deviation of the topographic height  $(\sigma_z)$  and two fitting

parameters ( $\sigma_0$  and  $\sigma_{mx}$ ), whose standard values are 100 and 1000  $\cdot \frac{64}{nlat}$  (where nlat is the number of grid cells in latitudinal direction considered in a setup) respectively.

$$b = \begin{cases} 0.5 & \text{if } \sigma_{mx} < \sigma_z \\ \frac{\sigma_z - \sigma_0}{\sigma_z - \sigma_{mx}} & \text{if } \sigma_0 < \sigma_z < \sigma_{mx} \\ 0.01 & \text{otherwise.} \end{cases}$$
 (A2)

Further assuming that the surface fraction on which runoff occurs  $(\frac{A_{Rsrf}}{A_{cell}})$  for a given grid-cell-mean water content is described by the above distribution function – that is  $\frac{A_{Rsrf}}{A_{cell}} = f_{(w)}$ , infiltration (I) can be determined as the residual of precipitation (P) – more specifically that fraction of precipitation that is not intercepted by vegetation, but including snow melt – and surface runoff ( $R_{srf}$ ):

$$\begin{split} I &= P - R_{srf} \\ R_{srf} &= P - \frac{(W_{rt,mx} - W_{rt})}{\Delta t} + \frac{1}{(b+1)^{b+1} \cdot W_{rt,mx}^b} \\ &\cdot \frac{1}{\Delta t} \left\{ \begin{bmatrix} (b+1) \cdot W_{rt,mx} \left(1 - \frac{W_{rt}}{W_{rt,mx}}\right)^{\frac{1}{1+b}} - P \cdot \Delta t \end{bmatrix}^{b+1} & \text{if } (b+1) \cdot W_{rt,mx} \left(1 - \frac{W_{rt}}{W_{rt,mx}}\right)^{\frac{1}{1+b}} > P \cdot \Delta t \\ 0 & \text{otherwise.} \end{split} \right. \end{split}$$

with  $W_{rt}$  being the water content of the root zone and  $W_{rt,mx}$  the maximum rootzone soil moisture.

Lateral drainage or subsurface runoff  $R_{blg}$  is calculated using two fixed drainage rates that are scaled by the saturation of the soil:

$$R_{blg} = \begin{cases} d_{mn} \cdot \frac{W_{rt}}{W_{rt,mx}} + (d_{mx} - d_{mn}) \cdot \left[ \frac{W_{rt} - W_{rt,crt}}{W_{rt,mx} - W_{rt,crt}} \right]^{d_{xp}} & \text{if } W_{rt} > W_{rt,crt} \\ d_{mn} \cdot \frac{W_{rt}}{W_{rt,mx}} & \text{otherwise} \end{cases}$$
 with

$$W_{rt,crt} = W_{rt,mx} \cdot f_{dr,crt}$$

where  $d_{mx}$  (2.81<sup>-8</sup>) corresponds to the maximum drainage that occurs under conditions close to saturation – that is for the soil moisture values about a critical threshold ( $W_{rt,mx} \cdot f_{dr,crt}$ , where  $f_{dr,crt} = 0.9$ ) and  $d_{mn}$  (2.81<sup>-10</sup>) to the maximum drainage rate in unsaturated soils.  $d_{xp}$  (1.5) is a drainage parameters used for model tuning. Assuming fully water saturated soils – i.e.  $R_{blg} = d_{mx}$  – the above parameter values allow for a volume of water corresponding to a typical excess water volume [i.e.  $W_{rt,mx} - W_{rt,crt}$  between 0.005 m and 0.03 m (Todini, 1996; Dümenil and Todini, 1992)], to drain from a catchment-sized area within 2.1 to 12.5 days.


In the second, the point scale, approach, soil moisture is assumed to be distributed evenly across the domain and the for-830 mation of surface runoff is exclusively based on the speed with which water can infiltrate into and move vertically through the soil. Here, infiltration (I) and surface runoff  $(R_{sfc})$  are determined in two steps, calculating the contribution of infiltration excess (Horton overland flow;  $R_{sfc,h}$ ) and saturation excess (Dunne overland flow;  $R_{sfc,d}$ ) separately. In the first step, the infiltration capacity,  $I_{cap}$  is calculated as the saturated hydraulic conductivity of the uppermost soil layer,  $k_{sat,1}$ , scaled by an ice impedance factor (Swenson et al., 2012),  $\Theta_{ice}$ :

$$I_{cap} = k_{sat} \Theta_{ice}$$
 with 
$$\Theta_{ice} = 10^{-6 \frac{\theta_{ice,1}}{\theta_{max,1}}},$$

where  $\theta_{ice,1}$  is the ice content of the uppermost soil layer and  $\theta_{max,1}$  the layer's porosity.

Horton overland flow,  $R_{sfc,h}$ , occurs if the available water exceeds the infiltration capacity and the surface storage capacity,  $I_{dpr}$ . Here, the model distinguishes between the water that can potentially infiltrate across the entire surface area  $I_{pot,sl}$  and the water,  $I_{pot,pnd}$ , that may additionally infiltrate within the inundated fraction  $fc_{pnd}$ . Consequently, the Horton overland flow consists of two elements, the infiltration excess that is generated across the entirety of the surface area,  $R_{srf,sl}^t$ , and the additional overflow of the surface depression storage  $R_{srf,pnd}^t$ :

$$R_{sfc,h} = R_{sfc,sl} + R_{sfc,pnd} \qquad \text{with} \qquad (A6)$$
 
$$R_{sfc,sl} = \begin{cases} I_{pot,sl} - I_{cap} & \text{if } I_{pot,sl} > I_{cap} \\ 0, & \text{otherwise} \end{cases} \qquad \text{and}$$
 
$$R_{sfc,pnd} = \begin{cases} I_{pot,pnd} - I_{cap} \cdot fc_{pnd} - I_{dpr}, & \text{if } I_{pot,pnd} > I_{cap} \cdot fc_{pnd} + I_{dpr} \\ 0, & \text{otherwise}. \end{cases}$$

Potential infiltration rates depend on precipitation P, or throughfall in vegetated areas, and on the water that is stored within surface depressions. Here, the assumption is being made that the precipitated water does not infiltrate uniformly across the surface area but predominantly pools within the depressions:

$$I_{pot,sl} = P \cdot (1 - fc_{pnd,mx}^{\frac{1}{3}})$$
 and 
$$I_{pot,pnd} = P \cdot fc_{pnd,mx}^{\frac{1}{3}} + \frac{W_{sfc}}{\Delta t},$$

where  $fc_{pnd,mx}$  constitutes the maximum inundated fraction and  $W_{sfc}$  is the current water content of the surface reservoir. The surface storage capacity is given by the maximum water content of the surface reservoir  $W_{sfc,max}$ :

$$I_{dpr} = \frac{W_{sfc,max}}{\Delta t}. (A8)$$

In the second step an upper limit for infiltration,  $I_{max}$ , is determined based on  $I_{pot,sl}$ ,  $I_{pot,pnd}$ ,  $I_{cap}$  and the inundated fraction  $fc_{pnd}$ :

$$I_{max} = \begin{cases} I_{pot,sl} + I_{max,pnd} & \text{if } I_{pot,sl} + I_{max,pnd} < I_{cap} \\ I_{cap} & \text{otherwise} \end{cases}$$
 with 
$$I_{max,pnd} = \begin{cases} I_{pot,pnd} & \text{if } I_{pot,pnd} < I_{cap} \cdot fc_{pnd} \\ I_{cap} \cdot fc_{pnd} & \text{otherwise.} \end{cases}$$

This rate is used as the upper boundary condition in the computation of the vertical soil hydrology. After the state of the soil column and the drainage fluxes have been updated any water that exceeds the pore volume is removed from the soil forming the saturation excess  $R_{sfc,d}$ . The latter is then used to estimate the actual infiltration and the total surface runoff:

$$\begin{split} I &= I_{max} - R_{sfc,d} & \text{and} & \text{(A10)} \\ R_{sfc} &= R_{sfc,h} + R_{sfc,d} & \text{with} \\ R_{srf,d} &= \sum_{i=1}^{nsoil} \frac{\theta^t_{upd,i} - \theta^t_{sat,i}}{\Delta t} & \text{for all } \theta^t_{upd,i} > \theta^t_{sat,i}. \end{split}$$

With respect to the subsurface lateral drainage, the ARNO-scheme determines the fluxes implicitly assuming a given retention time, to account for the temporal lag between runoff generation and the water reaching the closest tributary. For the point scale approach the generation of subsurface runoff was decoupled from these implicit assumptions and  $R_{lat}$  is instead determined as a function of the hydraulic conductivity,  $k_i$ , on a given layer i and the local slope m:

$$R_{blg,i} = k_i \cdot \sin\left(m \cdot \frac{pi}{2}\right),\tag{A11}$$

In addition to this lateral drainage component, we included the (vertical) drainage from the lowest soil layer b into the underlying bedrock,  $R_{bot,b}$ , with the drainage rates being limited by the hydraulic conductivity of the lowest layer,  $k_b$ , and the hydraulic conductivity of (fractured) bedrock,  $k_{rock}$ , assumed to be  $1 \cdot 10^{-7}$  m s<sup>-1</sup>.

$$R_{bot,b} = \begin{cases} k_b & \text{if } k_b 

In the vast majority of cases, the model is being used to represent areas that have a given spatial extent. This poses the problem, that any runoff determined by the above point-scale parametrizations can not be assumed to contribute to the stream flow instantaneously, since the water requires some time to move across a given distance either at the surface or through the ground before reaching a tributary. Here, the model makes use of intermediary reservoirs,  $L_{srf}$  and  $L_{blg}$  – analogues to those used in the retention parametrization in the hydrological discharge model – in which  $R_{srf}$  and  $R_{blg}$  are initially stored. The outflow of these reservoirs,  $R_{srf}^*$  and  $R_{blg}^*$ , is estimated based on assumed retention times  $\tau_{srf}$  and  $\tau_{blg}$  and constitutes the contribution to the streamflow.

$$R_{sfc}^* = \frac{L_{srf}}{\tau_{srf}}$$
 and 
$$R_{blg,i}^* = \frac{L_{blg,i}}{\tau_{blg}}.$$

For the surface runoff, infiltration or evaporation from the emerging flow channels are being neglected and the change in water content of the respective reservoirs,  $\Delta L_{srf}$ , is simply given by:

$$\frac{\Delta L_{srf}}{\Delta t} = R_{sfc} - R_{sfc}^*. \tag{A14}$$

The change in the water content on a given soil layer i of the reservoir retaining the lateral subsurface fluxes,  $L_{lat,i}$ , is calculated analogously, with the difference being, that the interactions between the intermediary reservoir and the surrounding ground need to be accounted for. Here, it is assumed that the lateral drainage ceases if the ground begins to dry out and all excess water has been removed from the soil. This behaviour is represented by a restoration flux  $R_{res,i}$ , which migrates water back into the soil matrix as soon as the soil water content,  $\theta_i$ , decreases below the soil field capacity,  $\theta_{fc,i}$ . Furthermore, the water within the intermediary reservoir may percolate downwards until it drains across the soil-bedrock interface, constituting additional bottom drainage,  $R_{bot,i}$ . We assume this flux to be limited by the hydraulic conductivity  $k_{rock}$  of (fractured) bedrock  $(1\cdot10^{-7}\,\mathrm{m\,s^{-1}})$  and the hydraulic conductivity,  $k_b$ , of the lowest soil layer above the bedrock boundary, b. Water is taken from a given layer of the intermediary reservoir until the volume of water that has been removed is equal to the bottom drainage flux multiplied by the time step length or until the intermediary reservoir is completely empty. This iterative process is started at the top of the soil column, and not at the soil-bedrock interface to account for the vertical movement within the reservoir. Here, it is assumed that the water percolates downward with the same rate as it drains into the bedrock:

$$\frac{\Delta L_{blg,i}}{\Delta t} = R_{blg,i} - R_{blg,i}^* - R_{res,i} - R_{bot,i} \qquad \text{with}$$

$$R_{res,i} = \begin{cases}
0, & \text{if } \theta_{fc,i} > \theta_i \\ \frac{\theta_{fc,i} - \theta_i}{\Delta t}, & \text{if } \theta_{fc,i} < \theta_i \text{ and } \theta_{fc,i} - \theta_i < L_{blg,i} \\ \frac{L_{blg,i}}{\Delta t} & \text{otherwise}
\end{cases}$$

$$\begin{cases}
0, & \text{otherwise} \\
\frac{L_{blg,i}}{\Delta t} & \text{otherwise}
\end{cases}$$

$$\frac{\Delta L_{blg,i}}{\Delta t} = R_{blg,i} - R_{blg,i}^* - R_{res,i} - R_{bot,i} \qquad \text{with}$$

$$R_{res,i} = \begin{cases} 0, & \text{if } \theta_{fc,i} > \theta_i \\ \frac{\theta_{fc,i} - \theta_i}{\Delta t}, & \text{if } \theta_{fc,i} < \theta_i \text{ and } \theta_{fc,i} - \theta_i < L_{blg,i} \\ \frac{L_{blg,i}}{\Delta t} & \text{otherwise} \end{cases}$$

$$R_{bot,i} = \begin{cases} 0. & \text{if } \frac{\sum_{i=1}^{i-1} L_{blg,i}}{\Delta t} > k_{bot} \\ \frac{L_{blg,i}}{\Delta t} & \text{if } \frac{\sum_{i=1}^{i} L_{blg,l}}{\Delta t}

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
