# Peer review of "T-REX: The tile-based representation of lateral exchange processes in ICON-Land"

_EGUsphere, 2025_

## Referee Comment (RC1)

The publication represents a strong showing on the potential for subgrid lateral exchange in a land surface model to significantly impact earth system models. The methods are described in great detail, with a few somewhat confusing areas that are understandable given the complexity of the issue. The paper would benefit from a bit more discussion on how this simplified tile interaction scheme would compare to higher resolution/more robustly physics based solutions to the problem. Additionally, the wind-blown snow section appears limited and has significant questions relating to the methodology. Excluding the section on snow redistribution, the publication is in a very strong shape and this reviewer looks forward to seeing it published.

**Major Comments**

1. In the sub-sections in 2.3, it would be nice to have a bit more discussion on how the simplifications would be expected to compare to a fully distributed model run at a higher resolution with more robust physics. I am not asking the authors to run a higher resolution to compare to, but rather would like some discussion on how the simplified equations here compare to more robust physics based solutions to these various fluxes (that may be computationally impractical), and/or to the core model simply run at a high enough resolution to consider these flows (and if this is even possible or if these represent processes that would be important even at higher spatial resolutions). This qualitative discussion would also open up an opportunity to highlight the computational benefit compared to what would be needed to fully represent these processes on a typical grid.

2. 2.2.1: Why are circles chosen as the characteristic shape as opposed to, say, rectangles where the shape can be preserved in the n-nested case? Rectangles also would follow the underlying grids informing the tile setup more naturally. Are there presumed issues with the rectangle format?

   The authors point out some potential issues with the choice of circles, namely the nested setup. I find the potential errors that result from different assumed shape to be concerning; could this be significant? If there is a valley, for example, would the different assumed shape depending on the connection create long term inaccuracies? Would it result in quicker or slower draining of the ridges to the valley? How big are these issues?

3. 2.3.3: I am not immediately convinced that the given approach would be appropriate for snow, whose redistribution will likely be driven substantially by wind which is excluded from both the scheme itself and the definition of the connectivity matrix (i.e. if the wind usually flows along the boundary between two tiles, the effective connectivity would be very low compared to a situation where the wind flows directly tile to tile).

   Is there any evidence that the redistribution of snow is simply elevation based? How strong is this assumption?

   What are the implications when it breaks down?

Is it a fair assumption to have it entirely independent of wind direction or magnitude?

More discussion is needed on this. If, for example, the dominant velocity is going upslope it is doubtful that snow would redistribute downslope. If velocities are very low, snow redistribution will be radically different from high velocities. Are tiles, and connectivity, designed for vegetation and subsurface flow the same tiles that would be chosen if snow redistribution was the primary concern? Considering the complex physics that should be at play for snow redistribution, and the importance of wind velocity both in magnitude and direction, and the difference in processes relative to subsurface lateral flow, I think at the least additional background information is necessary in this section to justify this decision, and significant discussion on the implications for complex terrain, different meteorological conditions etc.

The snow redistribution method is not seriously considered in the results either.

Overall, the snow part feels relatively poorly considered compared to the other sections on lateral connections for heat and liquid water and relies on very different fundamentals (i.e. atmospheric boundary layer becomes important, wind is important, obstructions such as vegetation become important) compared to heat/liquid water. This difference in fundamentals requires much deeper thought and discussion into snow that is not included here. I would encourage the authors to either be much more thorough in the discussion/determinations/limitations of snow processes or, likely better for a succinct publication, save this for a future publication where more background research, justification for the scheme, details on limitations, results focused on the impacts of this process and perhaps a scheme that considers velocity can be included.

**Minor Comments**

Figure 4 and lines 243-250: I find this a bit unclear. I understand the tile refined grid, but it definitely takes some work and staring at the image. I would consider trying to do this with a more simplified grid example. In particular, it is hard to understand what tile grid levels are interacting with what sibling grid levels.

Figure 4: it appears as though no water is flowing into the bottom level of the sibling native grid, if I am correct in interpreting the color scheme. This implies that the water in the tile would not flow along the bedrock. Does this introduce errors? Particularly in arid locations with a high elevation gradient? In an extreme case, one could even imagine a saturated zone with an unsaturated zone beneath it if horizontal flow is large, and subsurface is dry. Perhaps I am not understanding this correctly; I would appreciate clarification.

Line 65: Error in this phrase; "Max Planck Institute for Meteorology's- and the AWI Earth System … "

145-150: I find this confusing; are the authors saying that the connectivity matrix is then weighted by the area of the tile? Wouldn't this be problematic for, for example, a riparian zone tile which may be long and skinny and therefore have a small area but large contact perimeter?

Lines 580 - 581: A few too many dashes used in place of commas or other separation words which makes it a bit confusing

Does the subgrid scheme you have implemented increase or decrease subgrid variability in heat/moisture fluxes and soil moisture? We see this for temperature, but not for moisture or the surface fluxes (perhaps I missed this). Does this, qualitatively, match the expectations from remote sensing etc.? I ask particularly in reference to land-atmosphere interactions (which, given one way coupling, is not fully considered here). If subgrid variability is increased, as I suspect, does this pose problems for the typical flux homogenization (flux averaging/aggregation) done between the land and atmosphere given increasing work/emphasis on considering subgrid atmospheric affects driven by surface heterogeneity, such as secondary circulations? (Waterman 2025, Arnold 2024, Fowler 2024, Huang 2022) These questions cannot be fully answered with this analysis, and is not expected to be addressed, but has interesting applications in this area facing growing interest.